# Biogeographic gradients of picoplankton diversity indicate increasing dominance of prokaryotes in warmer Arctic fjords

Cora Hörstmann [1,2,3] ✉, Tore Hattermann [4,5], Pauline C. Thomé[6], Pier Luigi Buttigieg[7], Isidora Morel[8], Anya M. Waite[9] & Uwe John[1,10]

Climate change is opening the Arctic Ocean to increasing human impact and ecosystem changes. Arctic fjords, the region's most productive ecosystems, are sustained by a diverse microbial community at the base of the food web. Here we show that Arctic fjords become more prokaryotic in the picoplankton (0.2–3 μm) with increasing water temperatures. Across 21 fjords, we found that Arctic fjords had proportionally more trophically diverse (autotrophic, mixotrophic, and heterotrophic) picoeukaryotes, while subarctic and temperate fjords had relatively more diverse prokaryotic trophic groups. Modeled oceanographic connectivity between fjords suggested that transport alone would create a smooth gradient in beta diversity largely following the North Atlantic Current and East Greenland Current. Deviations from this suggested that picoeukaryotes had some strong regional patterns in beta diversity that reduced the effect of oceanographic connectivity, while prokaryotes were mainly stopped in their dispersal if strong temperature differences between sites were present. Fjords located in high Arctic regions also generally had very low prokaryotic alpha diversity. Ultimately, warming of Arctic fjords could induce a fundamental shift from more trophic diverse eukaryotic- to prokaryotic-dominated communities, with profound implications for Arctic ecosystem dynamics including their productivity patterns.

Arctic fjords are among the most productive high-latitude regions of the ocean's biosphere, sustaining important fisheries[1]. Due to increasing anthropogenic climate change, fjords have been subject to severe ecosystem changes such as glacial retreat, changes in freshwater input, and altered matter exchange between terrestrial and coastal ocean systems[2]. Fjords are generally marked by strong land-sea interactions and span a large range of bioclimatic subzones, including temperate, subarctic, and Arctic regions. Arctic fjords are characterized by steep spatial and temporal environmental gradients, including strong seasonality[3] and physical gradients along the fjord length, which are driven by dynamic inputs of freshwater and nutrient release from glaciers and river runoff, resulting in highly complex

ecosystems[4,5]. Environmental changes in response to anthropogenic perturbations and climate change pose a serious threat to local biodiversity and fjord ecosystem function[6].

Microorganisms, the key players in marine primary productivity and recycling organic matter, form the base of the marine food web and drive major biogeochemical cycles such as carbon and nitrogen[7]. They form unique, regional communities in sub-Antarctic fjords[8], whose occurrence and/or maintenance is of high interest in understanding the coupling between biodiversity and physical dynamics of Arctic ecosystems[9]. Analysis of microbes and their role in Arctic and sub-Arctic ecosystems requires trait-based approaches, especially considering the large size range (micro-, nano-,

[1]Alfred Wegener Institute Helmholtz Center for Polar and Marine Research, Am Handelshafen 12, 27570 Bremerhaven, Germany. [2]Aix Marseille Univ, Universite de Toulon, CNRS, IRD, MIO UM 110, 13288 Marseille, France. [3]Turing Center for Living Systems, Aix-Marseille University, 13009 Marseille, France. [4]Norwegian Polar Institute, iC3: Centre for Ice, Cryosphere, Carbon and Climate, Framsenteret, Hjalmar Johansens gate 14, 9296 Tromsø, Norway. [5]Complex Systems Group, Department of Mathematics and Statistics, The Arctic University – University of Tromsø, Hansine Hansens veg 18, 9019 Tromsø, Norway. [6]Leibniz Institute of Freshwater Ecology and Inland Fisheries, Müggelseedamm 310, 12587 Berlin, Germany. [7]Helmholtz Metadata Collaboration, GEOMAR, Wischhofstraße 1-3, 24148 Kiel, Germany. [8]Max Planck Institute for Marine Microbiology, Celsiusstraße 1, 28359 Bremen, Germany. [9]Ocean Frontier Institute, Dalhousie University, 1355 Oxford Street, Halifax, NS, Canada. [10]Helmholtz Institute for Functional Marine Biodiversity at the University of Oldenburg (HIFMB), Ammerländer Heerstraße 231, 26129 Oldenburg, Germany. ✉e-mail: cora.hoerstmann@awi.de

and picoplankton) and differential ecological roles of microbes[10]. The smallest of the planktonic size classes, the picoplankton, include picoeukaryotes and prokaryotes (both defined by a cell size of 0.2–3 μm), who share several traits and together make up a large fraction of the photoautotrophic biomass[11]. There are, however, large discrepancies between the environmental factors, e.g., temperature and nutrients, driving prokaryotic and picoeukaryotic distribution[12]. The impact of regional selection processes on Arctic picoplankton diversity is incompletely understood, limiting the ability to project how environmental changes will impact fjord ecosystems.

Anthropogenic climate change has regionally increased heat transport through poleward extension of Atlantic water, referred to as Atlantification, altering the 'trails of life and death' of species into the Arctic Ocean[13]. Specifically, this has promoted the invasion of more boreal species into the Arctic Ocean with potential alterations in energy transfer to higher trophic levels[14]. This also includes a northward shift of key planktonic species in marine biogeochemical cycles, such as *Emiliania huxleyi*[15,16] and *Phaeocystis pouchetii*[17]. Considering that temperature is a main driving factor for prokaryotic diversity[18], increasing temperatures within the Arctic Ocean may particularly alter prokaryotic diversity in comparison to picoeukaryotes. Moreover, warmer water temperatures can increase the metabolic rates of microorganisms, ultimately increasing primary productivity[19] and altering the local carbon pool[20]. Detailed ecological studies at multiple sites with different oceanographic and environmental conditions are needed to elucidate the complex climate-related changes in microbial diversity and associated changes in food webs and carbon cycling.

Here, we investigated the impact of oceanographic connectivity and local selection process on picoplankton communities of multiple Arctic, subarctic, and temperate fjords, which included samples from 93 sites in 21 fjords in Sweden/southern Norway, northern Norway, Svalbard, Iceland, East Greenland, and West Greenland. We focus on picoplankton size fraction only, as picoplankton are likely to play an increasingly important role in the Arctic Ocean in the future due to their high nutrient affinity[11], large effective population size, and high genetic diversity[21]. We hypothesize that changes in picoplankton community compositions (beta diversity) would be related to regional and pan-Arctic oceanographic transport of microorganisms, while being limited by local environmental selection processes[13]. To identify these regional drivers, we applied multidimensional analyses of prokaryotic and eukaryotic 16 S and 18 S rRNA gene sequences, together with environmental variables and trophic classifications of prokaryotic and eukaryotic taxa. Further, we related changes in picoplankton beta diversity to modeled oceanographic connectivity at the basin scale. We expect different regionally constrained signals for prokaryotes and picoeukaryotes, following the assumption that prokaryotes and picoeukaryotes are intrinsically differently constrained in their local selection by, for example, temperature[22], light availability[23], and nutrient input dynamics[24]. These different responses to environmental changes can have cascading effects on the trophic structure within the entire picoplankton community[25]. Our analysis contributes to an improved understanding of the microbial ecosystem structure on a pan-Arctic scale in an ecologically-sensitive region subject to rapid and dramatic climate-driven changes.

## Results
### Geographic location, regional partitioning, and oceanographic connectivity
We analyzed a total of 93 surface ocean samples from Arctic, subarctic, and temperate fjords between 2012 and 2019 in the northern hemisphere spring and summer. Three to six samples were taken along the length of each fjord (Figs. S1–S6). Between fjords, the samples were clustered in six geographically distinct regions: northern Norway, Sweden/southern Norway, Svalbard, Iceland, East Greenland, and West Greenland (Fig. 1a, Supplementary Data 1). The regions can be further attributed to four distinct bioclimatic subzones as defined by the Circumpolar Arctic Vegetation Mapping Project (https://www.arcticcentre.org/EN/arcticregion/Maps/definitions). These regions are oceanographically influenced by cold and low-salinity polar water and warmer and more saline Atlantic water (Fig. S7).

We found significant differences in water mass characteristics between bioclimatic subzones (Kruskal–Wallis, $p < 0.001$), but also high regional variability in dissolved inorganic nutrient ($NO_3$, $PO_4$, and Si) concentrations (Fig. 1b), primarily due to relatively higher nutrient concentrations in the fjord heads in comparison to other samples in the fjord central part and mouths (Supplementary Data 1). Due to the high internal variability, we could not identify single environmental variables indicative of temperate, subarctic, or Arctic regions (Fig. 1b).

Trajectories of Lagrangian drifters released for ≤ 3 months exhibited strong regionality (e.g., only fjords within northern Norway were interconnected). The primary trajectory pathways at > 6 months were (1) from southern Norway to northern Norway along the North Atlantic Current; (2) from northern Norway along the Barents Sea towards west of Svalbard, and from Svalbard (and Iceland) along the east Greenland Current towards west Greenland (Fig. 1c, Figs. S8–S14). At the maximum simulation time of 5 years, drifters exhibited a widely dispersed distribution across the Arctic Ocean, with Svalbard, east and west Greenland fjords receiving drifters from Sweden/southern Norway, northern Norway, and Iceland (Fig. S13).

### Picoplankton beta diversity structure and function
We found in total 6564 picoeukaryotic and 3899 prokaryotic ASVs, whose differential abundances between sites (beta diversity) correspond to biogeographic and ecological regions. Specifically, community similarity was higher within than between bioclimatic subzones. It intensified even more in the increasingly smaller biogeographic units of geographic regions and individual fjords according to redundancy analyses (RDA) (Fig. 2a, b). This effect was also significant in permutational ANOVA (PERMANOVA of bioclimatic subzone: prokaryotes, $R^2 = 0.4$, $p = 0.001$; eukaryotes, $R^2 = 0.4$, $p = 0.001$; PERMANOVA of geographic regions: prokaryotes, $R^2 = 0.5$, $p = 0.001$; eukaryotes, $R^2 = 0.6$, $p = 0.001$; PERMANOVA of individual fjords: prokaryotes, $R^2 = 0.7$, $p = 0.001$; eukaryotes, $R^2 = 0.7$ $p = 0.001$). Sites from East Greenland ordinated farther away than other samples in the eukaryotic RDA (Fig. 2a). We found one Station (MSM21.F02.514) in the regional groupings from Disko Bay that ordinated closer to samples from northern Norway than other sites from West Greenland. Within prokaryotes, we found an outlier (MSM56.F02.553) that ordinated closer to samples from Iceland than to other samples from Kongsfjorden (Svalbard).

We noted that patterns among picoeukaryotes and prokaryotes varied in their spread of beta diversity patterns within individual bioclimatic and geographic regions. Among eukaryotes, beta diversity ordinated evenly between bioclimatic subregions. Sites separated into high Arctic, low Arctic, subarctic, and temperate clusters along the first RDA axis, which captured 46.53% of the variation. Additionally, sites in East Greenland (low Arctic) clearly separated from other low Arctic sites (West Greenland) along the second RDA axis (Fig. 2a). Prokaryotic beta diversity had strong differences between bioclimatic subregions with higher spread in temperate regions in comparison to low and high Arctic regions (Fig. 2b). Specifically, ordination of prokaryotic beta diversity sites showed an exceptionally high spread among temperate sites. Conversely, all Arctic regions primarily overlapped with each other (Fig. 2b), forming one general Arctic beta diversity signal.

Between Arctic, subarctic, and temperate regions, microbial beta diversity could be explained by different concentrations of nutrients ($NO_{3-}$, $PO_4^{3-}$), light availability (approximated by sun elevation angle), depth, temperature, and salinity. Among these variables, temperature stood out as the most important structuring environmental variable (Table S1) that also aligned with the first (and most explanatory) RDA axis in both eukaryotes and prokaryotes. In our eukaryotic RDA, almost all environmental variables (temperature, bottom depth, fluorescence, $NO_{3-}$, and $PO_4^{3-}$), except sun elevation angle, salinity, and presence of silicate, were positively associated with subarctic/temperate regions (Fig. 2a, b). Silicate was positively associated with the first RDA axis in the eukaryotic analysis, and thus predominately positively associated with more Arctic-influenced sites, which also exhibit marine-terminating glaciers (Fig. 2a, Supplementary Data 1). In our prokaryotic RDA, silicate and fluorescence had no significant association (stepwise permutation model analysis). Sites belonging to temperate

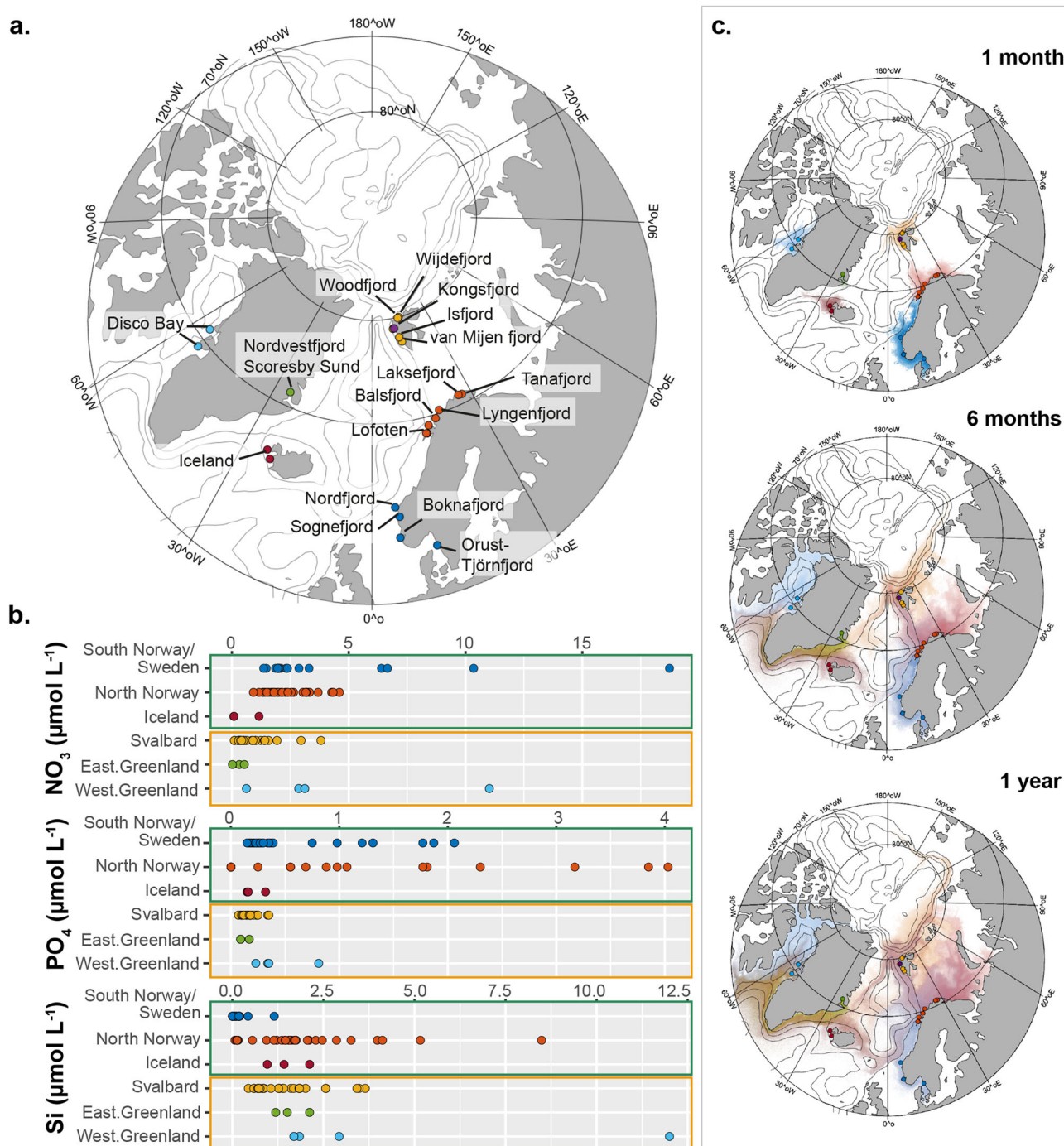

**Fig. 1 | Oceanographic connectivity. a** Map of fjords. Sample sites, color-coded to different regions. **b** Dissolved inorganic nutrient concentrations of nitrate (NO₃), phosphate (PO₄), and silicate (Si) in μmol L$^{-1}$ in Sweden/southern Norway ($n = 17$), northern Norway ($n = 30$), Iceland ($n = 3$), Svalbard ($n = 27$), eastern Greenland ($n = 3$), and western Greenland ($n = 4$). Sites are color-coded as in (**a**), dark green boxes indicate regions without marine-terminating glaciers, orange boxes indicate regions with marine-terminating glaciers. **c** Trajectories of modeled drifters mapping oceanographic connectivity of individual fjords after 1 month, 6 months, and one year (see Figs. S2–S7 for connectivity matrices). See Fig. S10 for full tracking (up to 5 years). Sites and drifters are color-coded according to geographic regions of release where darker colors indicate higher drifter concentrations.

regions, and thus more influenced by Atlantic water which is generally considered warmer and nutrient-rich, were strongly associated with temperature, NO₃₋, and PO₄³⁻ concentrations (Fig. 2b). Notably, temperature captured two times more of the variance for prokaryotes than for eukaryotes (PERMANOVA, 38% for prokaryotes vs. 19% for eukaryotes, Table S1).

Collapsing beta diversity signals to differences in temperature only, we found that Aitchison distances correlated with temperature differences (Δ temperature) between sites for eukaryotes and prokaryotes (Fig. 2c, d).

Prokaryotic Aitchison distance was significantly correlated with differences in temperature between all bioclimatic subzones. In contrast, eukaryotic Aitchison distance was significantly correlated, but confidence intervals were wide with temperature differences, especially when crossing from temperate into subarctic regions (Table S2).

Resembling the overall beta diversity patterns, prokaryotic richness was significantly lower in [cold] Arctic regions (adjusted $p < 0.001$, $n_{highArctic} = 33$, $n_{lowArctic} = 10$, $n_{subarctic} = 37$, $n_{temperate} = 18$; Fig. 3a, Table S3). Temperate

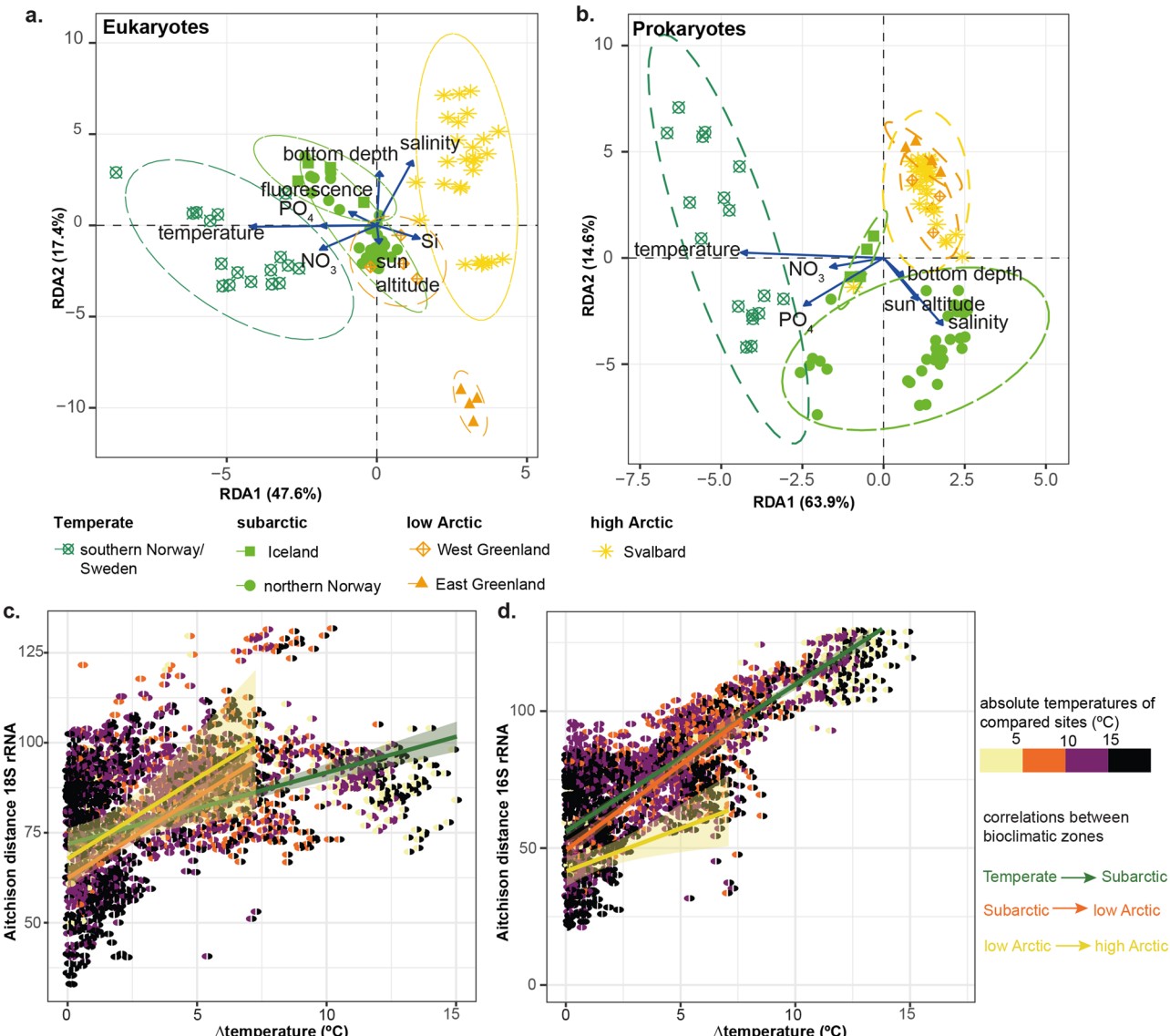

**Fig. 2 | Picoplankton beta diversity distribution. a** Redundancy analysis (RDA) of CLR-transformed eukaryotic ASV table (30% of the variance in the ASV table constrained, $p = 0.001$, permutations = 999, $n = 90$); **b** RDA of CLR-transformed prokaryotic ASV table (43% of the variance in the ASV table constrained, $p = 0.001$, permutations = 999, $n = 93$). Colors and shape corresponds with geographic regions; **c** Aitchison distance of picoeukaryotic community composition against temperature differences between sites. Each site pair is color coded according to the absolute temperatures of each site with the site with lower temperature in the left half-circle and the site with the higher temperature in the right half-circle. **d** Aitchison distance of prokaryotic community composition against temperature differences between sites.

regions had generally significantly higher richness and evenness among both eukaryotes and prokaryotes (Fig. 3, Table S3). Among all bioclimatic subzones, the subarctic sites were the only ones with a higher prokaryotic richness ($560 \pm 22$, median ± s.e.m, $n = 37$) than eukaryotic richness ($444 \pm 31$, median ± s.e.m, $n = 39$). Despite changes in alpha and beta diversity across regions, relative abundances of major taxonomic groups did not majorly change at the Order level across bioclimatic subzones (Fig. S15).

We tested the maintenance of trophic functions across bioclimatic subzones by trophic trait assignment, which infers the relative contributions of individual trophic groups based on ASV abundance. Notably, trait-based analysis does not account for absolute gene copy numbers per cell variability, which is usually normalized in functional inference algorithms such as PICRUST2[26]. However, this relationship is currently incompletely understood for eukaryotes. Therefore, we applied trait-based analysis to compare prokaryotes and eukaryotes. As an example of the intercomparability between the two approaches, we mapped our trophic annotation against the functional inference of prokaryotes for "autotrophy" including

photoautotrophs and chemolithoautotrophs. ASV abundance of autotrophic prokaryotes correlated well with gene copy numbers involved in autotrophy derived from PICRUST2 (Fig. S16).

The relative functional contribution of picoeukaryotes and prokaryotes significantly differed between all bioclimatic subzones (with high Arctic and low Arctic merged due to low sample size; Fig. 4; Table S4). Specifically, Arctic sites had significantly more autotrophic and mixotrophic eukaryotes than subarctic and temperate regions (adjusted $p < 0.001$, Table S4). Additionally, we found significantly more heterotrophic eukaryotes in Arctic and temperate regions than in subarctic regions (adjusted $p < 0.001$), but no significant difference between Arctic and temperate regions (adjusted $p = 0.3$; Table S4). Conversely, subarctic and temperate sites had significantly more heterotrophic bacteria than Arctic sites (Arctic – subarctic: adjusted $p < 0.001$; Arctic – temperate: adjusted $p < 0.03$). The subarctic samples had proportionally more prokaryotes than the Arctic and temperate regions, which was also significant in all functional groups (autotrophic, mixotrophic, and heterotrophic; Fig. 4, Table S4, Fig. S17).

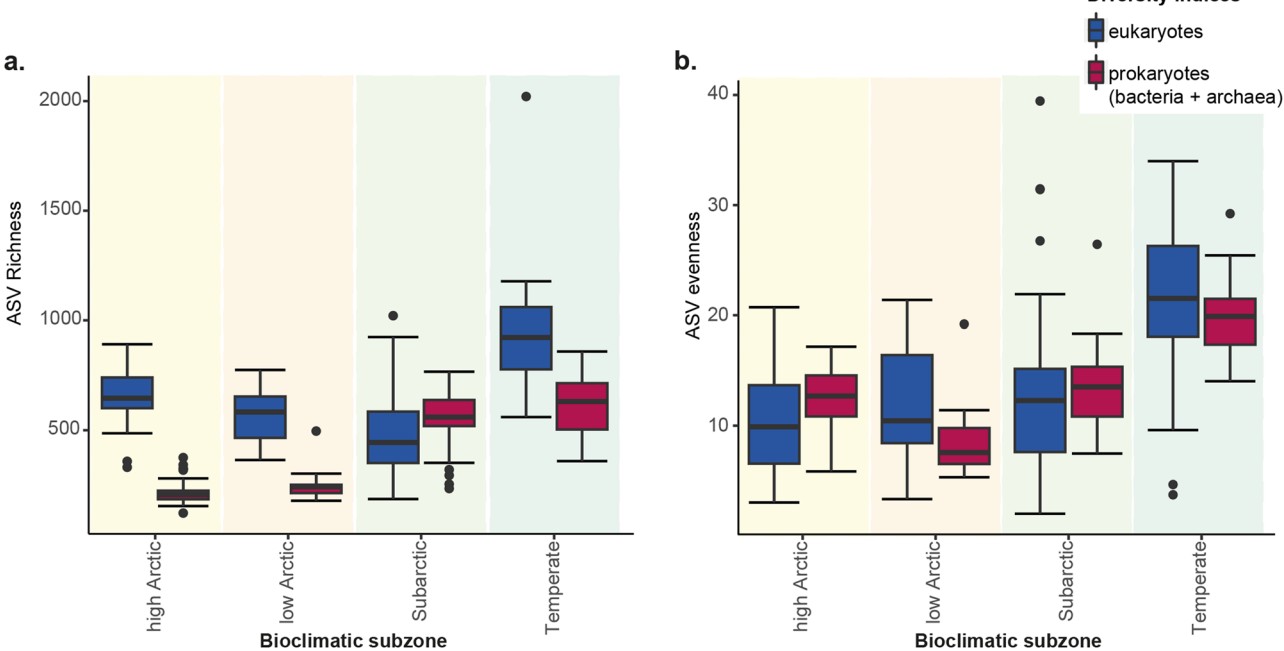

**Fig. 3 | Alpha diversity scores in bioclimatic subregions. a** Boxplot of ASV Richness within each bioclimatic subzones indicating median, upper and lower hinges, whiskers and outliers. **b** Boxplot of ASV Pielou Evenness within each bioclimatic subzones indicating median, upper and lower hinges, whiskers and outliers.

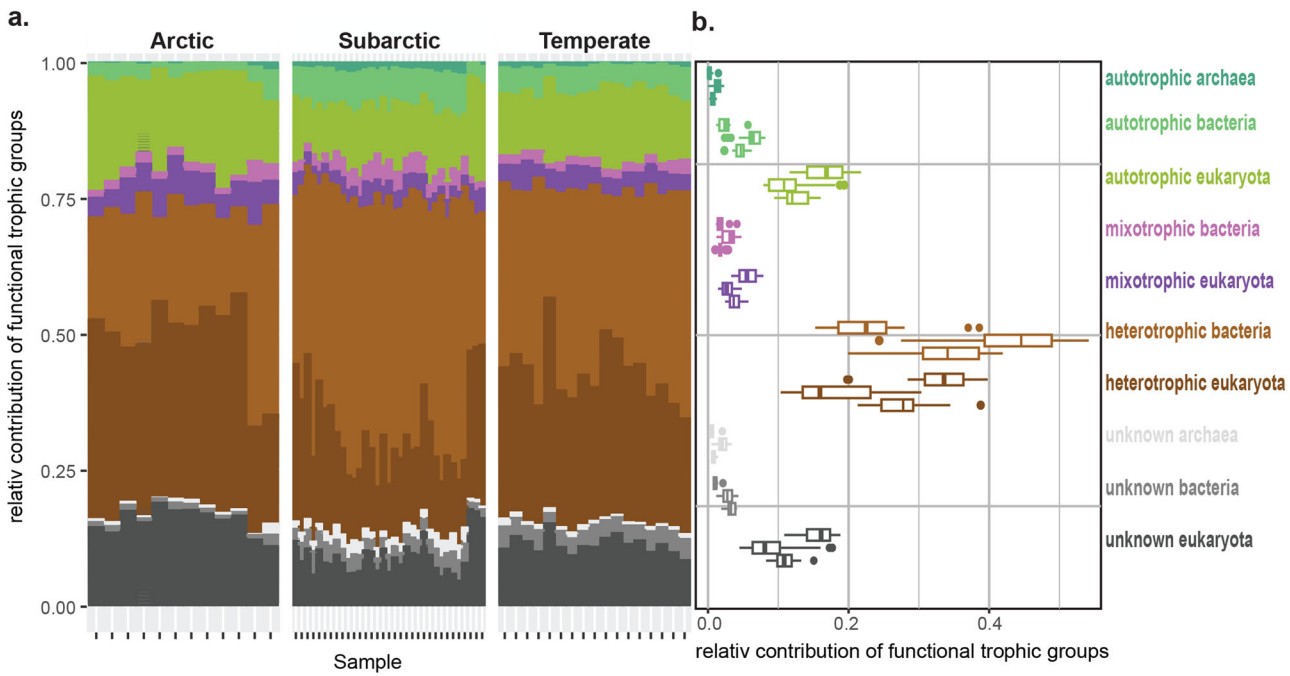

**Fig. 4 | Picoplankton trophic functional groups. a** Relative contribution of trophic functional groups of picoeukaryotes and prokaryotes of Arctic, subarctic and temperate regions. Annotated taxa are summarized in Supplementary Data 2. CLR-transformed ASV tables of picoeukaryotes and prokaryotes were merged and normalized to 1. **b** sum of each trophic functional group within bioclimatic subzones showing medians, upper and lower hinges, whiskers and outliers. For each trophic functional groups, the boxplots are ordered from top to bottom: Arctic – subarctic – temperate.

## Oceanographic connectivity

For each fjord, picoplankton beta diversity distance (Aitchison distance) of the sites closest to the fjord mouth was compared to the respective hydrodynamic distance (defined as the normalized inverse of the log10 of the synthetic particle concentrations) of connected site pairs. Sites were considered "representative sites" of individual fjords as between-fjord beta diversity structure was significantly different in comparison to within-fjord beta diversity (PERMANOVA, prokaryotes, $R^2 = 0.7$, $p = 0.001$; eukaryotes,

$R^2 = 0.7$, $p = 0.001$). Hydrodynamic distance was positively correlated with pro- and eukaryotic Aitchison distances across all temporal bins (Fig. 5; Pearson correlation), reflecting similarities of microbial communities that are oceanographically connected. Sites that were revisited over multiple years (Lofoten/Vesterålen (northern Norway) in 2014 and 2019; Kongsfjorden (Svalbard) in 2016 and 2017) were clear outliers relative to other sites with similar hydrodynamic distance, particularly in the 1-, 3-, and 6-month bins (Figs. S14 and S18).

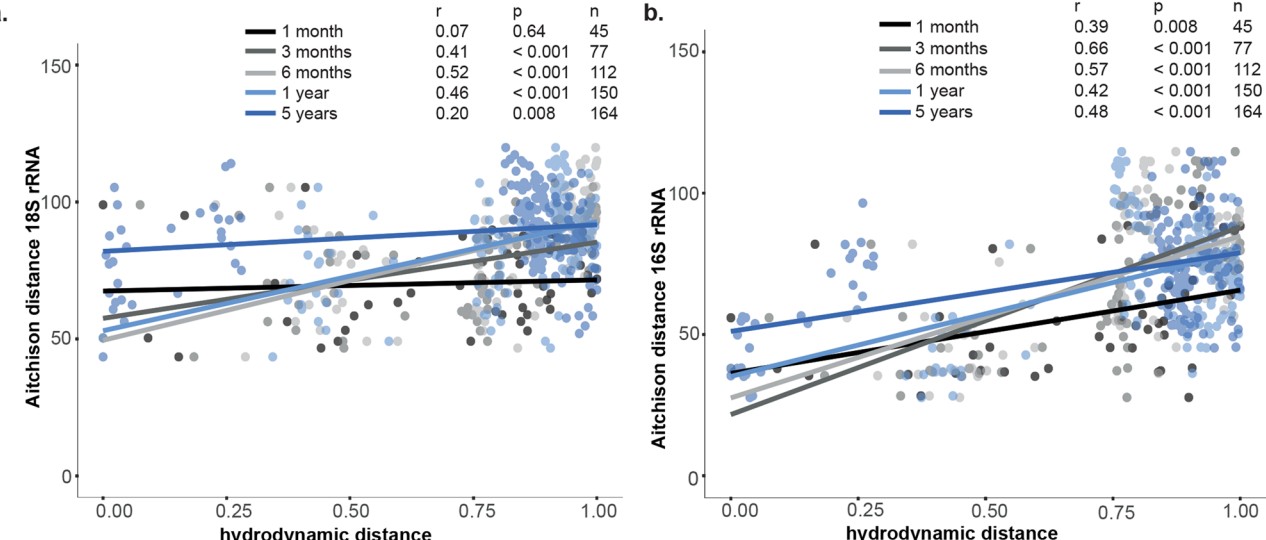

**Fig. 5 | Oceanographic connectivity between sites. a** Aitchison distance analysis of picoeukaryotic community composition (18 S rRNA sequences) against hydrodynamic distance defined as the inverse of the $\log_{10}$ of the synthetic drifter concentration normalized to the range from 0 to 1. Correlations are color-coded and Pearson correlations calculated for each temporal bin. Sites without oceanographic connection are not included; **b** Aitchison distance analysis of prokaryotic microbial communities (16 S rRNA sequences) against hydrodynamic distance defined as the inverse of the $\log_{10}$ of the synthetic drifter concentration normalized to the range from 0 to 1. Correlations are color-coded and Pearson correlations calculated for each temporal bin. Sites without oceanographic connection are not included.

We observed a slightly stronger correlation for prokaryotes with hydrodynamic distance than for eukaryotes (Fig. 5). Eukaryotes exhibited high internal variability of Aitchison distances at sites with a comparable degree of hydrodynamic distance (Fig. 5a). This variability also increased with differences in temporal bins. Among prokaryotes, Aitchison distances between sites from Sweden/southern Norway (temperate regions) to northern Norway (subarctic region) were more pronounced than other sites with comparable hydrodynamic distances that also had less temperature differences between sites (Fig. 3). Contrastingly, the Aitchison distance was comparably small between sites from Svalbard and East Greenland, as well as between sites from Svalbard and West Greenland (Arctic regions). Overall, prokaryotic and eukaryotic Aitchison distances were stronger when crossing bioclimatic subzones (i.e., temperate to subarctic to Arctic) than within bioclimatic subzones despite comparable oceanographic connectivity (Fig. S18).

## Discussion

The presented study revealed oceanographic connectivity of microbial communities within and between temperate, subarctic, and Arctic regions through oceanographic transport. However, this relationship was weakened when crossing between major bioclimatic subzones, particularly among prokaryotes (temperate to subarctic region; drifter time 3–6 months). Instead, we found a strong correlation between temperature differences and differences in picoplankton community composition (Aitchison distance), particularly within prokaryotes, indicating temperature barriers for prokaryotic dispersal into Arctic regions. Indeed, only few prokaryotic taxa thrive in Arctic fjords, which was reflected by low prokaryotic Richness. We documented significantly, proportionally more relative abundances of eukaryotic auto-, mixo- and heterotrophs in Arctic fjords, with subarctic and temperate fjords comprising relatively more prokaryotic auto- and heterotrophs. We demonstrated that picoplankton beta diversity separated statistically into high Arctic, low Arctic, subarctic, and temperate fjords, arising from different relative abundances at the ASV level, which were, however, not reflected at higher taxonomic ranks (Order level or higher). These regions were also distinct in their environmental conditions, such as nutrient profiles, with temperature being the most important structuring variable between sites. As temperature differences may decline in a future Arctic Ocean[27], reduced temperature barriers can support more prokaryotic expansion into Arctic regions with fundamental ecosystem baseline shifts among trophic functional groups.

Advection is a key determinant of microbial survival in the Arctic Ocean but is yet inadequately quantified at scales relevant for biogeochemical cycles and species distribution[13]. Our drifter analysis revealed strong regionality among shorter timeframes (< 6 months) and a nearly pan-Arctic distribution across multiple years (but no recirculation of northern sites into southern fjords in southern Norway or Iceland, which requires time scales >10 years). Changes in microbial community composition (Aitchison distance) positively correlated with hydrodynamic distance, supporting previous observations of structuring importance of hydrodynamics for microbial diversity assemblies[28]. Consistent with these findings, the Nordvestfjord Scoresby Sund (East Greenland), a region strongly affected by the cold East Greenland Current (low temperature/low salinity; Fig. S7) and weakly connected to other sites in our data, clearly ordinated from other sites along the second RDA axis in eukaryotic picoplankton.

Prokaryotes were more influenced by hydrodynamic distance than were eukaryotes, suggesting that prokaryotes were more driven by geographic dispersal[12]. However, prokaryotic beta diversity was even stronger associated with temperature differences between sites, which could not only be explained by temperature being a water mass indicator, suggesting ecological temperature barriers for prokaryotes. Additionally, we found significantly fewer prokaryotic ASVs in the [cold] Arctic regions. Further, prokaryotic beta diversity distance was strongest between the relatively warm fjords in Sweden/southern Norway and relatively cold fjords in northern Norway (up to 15.01°C temperature differences between sites) despite their relatively strong oceanographic connectivity. This suggests that temperature is an important ecological structuring variable for prokaryotes, currently limiting many prokaryotic taxa to thrive in Arctic fjords. However, prokaryotic dispersal could increase in the future due to increasing Atlantification of the Arctic Ocean[29], with even more rapid ecological changes through local temperature-driven stimulation of dormant stages in both prokaryotes and eukaryotes[30,31]. Such temperature shifts will lead to expansions of suitable habitats to thrive, as observed for the eukaryotic nanoflagellate *Emiliania huxleyi* over the past 24 years[15] and the shift of diatom-dominated spring communities in the Fram Strait to the domination of *Phaeocystis pouchetii*[17] and their strong competitive advantage in Arctic coastal regions[32]. So far, previous studies focused on the effect of Atlantification on eukaryotic key species only. Our comparative analysis of picoeukaryotic and prokaryotic dispersal patterns highlights a potentially

emerging dominance of more diverse prokaryotes in previously eukaryotic-dominated Arctic ecosystems within the picoplankton size fraction, adding a holistic perspective of future ecosystem changes.

Prokaryotes were less diverse in Arctic regions than eukaryotes, with largely overlapping beta diversity in low Arctic and high Arctic fjords, likely due to a lower competitiveness compared to picoeukaryotes. However, notably, we found the highest proportion of prokaryotes in subarctic fjords despite apparent temperature barriers between temperate and subarctic regions. We could not link this observation to any measured variable. This highlights the limitation of our current knowledge on these rapidly changing ecosystems and the urgency to study them more multidisciplinary, and to extend future studies by including other ecological variables, such as the effect of biological interactions (grazing, symbiosis, biological dispersal) that span across size classes[33].

Although observations across temporal scales were not the primary objective of our analysis, we detected both inter-annual and seasonal effects in beta diversity signals, likely because the dataset spanned across multiple years (2012–2019) and ranged seasonally from the end of May to mid-August. For example, the Lofoten/Vesterålen (northern Norway) and Kongsfjorden (Svalbard) sites that were repeatedly sampled in our dataset exhibited strong temporal changes in the picoplankton diversity despite their regionality, suggesting inter-annual and/or seasonal variabilities. In summer, glacial-influenced regions in the low Arctic and high Arctic bioclimatic subzones can be shaped by strong stratification through differences in ice cover and freshwater influence from melting glaciers and permafrost[34]. Similarly, seasonal shifts in microbial relationships have been observed between a parasitic network during the winter and a detritus-based food web in summer along the Alaskan Beaufort Sea coast[35] and alterations within the food web, such as episodic events of rapidly sinking ice algae[36]. At present, we cannot exclude a seasonal effect induced through the different geographic (latitudinal) locations on the observed pattern in our data, which could be related to diverging seasonal signals, such as different timing of ice melt and river discharge[37]. To rule out temporal variabilities, a possible sampling design could include a set of samples taken at the same time point (comparable to the Ocean Sampling Day[38]) and scattered across multiple fjords and analyzed combined with particle models to identify water masses of origin.

Fjord systems form unique and productive habitats for marine life, and their distinct microbial biodiversity patterns in relation to the presence of marine-terminating glaciers have been recently highlighted in the Southern Hemisphere across the Patagonian fjords[8]. Our observations confirmed that these distinct diversity patterns also exist in the picoplankton size fraction between bioclimatic subzones. Moreover, the beta diversity patterns resemble the recently observed breakpoints in beta diversity patterns of temperate vs. polar algal microbiomes[39], which are driven by distinct metabolic adaptations to changes in temperature and salinity[40]. While we could link prokaryotic diversity to temperature barriers, we generally observed a more hierarchical organization of spatial levels of microbial diversity patterns (bioclimatic subzones – geographic regions – fjords) instead of beta diversity breakpoints. The nuanced differences between the beta diversity patterns of picoeukaryotes and prokaryotes highlight different environmental drivers and adaptation strategies in the different taxonomic groups, which should be considered when studying environmental changes within and between these different ecosystem levels.

Arctic picoeukaryotic beta diversity was positively associated with Si concentration, a key element for diatoms, whose concentrations will change with glacial melt and increased acidification in Arctic bioclimatic subzones, resulting in cascading effects on the microbial community[41]. Other than Si, we could not determine any environmental parameters, for example, increased organic matter and nutrient input from glacier runoffs associated with the environmental conditions of Arctic fjords[4,5]. Furthermore, we did not detect a significant increase of prokaryotic taxa involved in iron and sulfur cycling (*Sulfitobacter* spp., Thiotrichales, Thiomicrospirales), which have previously been associated with glacial-influenced fjord systems (Van Keulenfjorden, Svalbard[42]; Kongsfjorden, Lilliehöökfjorden, Dicksonfjorden, Svalbard[43]). This lack of iron and sulfur cycling taxa could be due to seasonal

variations in the magnitude of iron and sulfur flux and location[44] and/or comparable input through enriched sediments in non-glacial-influenced fjords, such as in the Norwegian shelf[45,46]. We did observe an outlier in the eukaryotic RDA, in which the station closest to the coast in Disko Bay ordinated closer to stations from northern Norway and had much higher N and P concentrations than other samples from Disko Bay, potentially arising from glacier melt and associated nutrient input[47]. Overall, this suggests a more complex interplay of microbial diversity, and regional environmental conditions, such as nutrient input, changes in light availability, and stratification, which compromises the ability to conclude a general shift in ecosystem structure on a pan-Arctic scale[2].

Recent work indicates that higher temperatures in the Arctic can increase bacterial production relative to viral lysis, resulting in more efficient cycling of bacterial carbon within the microbial loop[22]. Additionally, a positive impact of glacier melt on heterotrophic metabolism of prokaryotes has recently been highlighted in the Antarctic Getz Ice Shelf region[48]. Our observations support and extend these observations: We observed proportionally more eukaryotic functional groups in Arctic fjords with proportionally more heterotrophic prokaryotes in subarctic and temperate fjords. Notably, the ratios between functional groups (auto-, mixo-, heterotroph) remained constant throughout all samples, but the fraction of prokaryotes and picoeukaryotes within these functional groups changed significantly between regions. This suggests that picoeukaryotes and prokaryotes can serve the same function within an ecosystem, a form of functional redundancy that actually occurs between domains. It remains unclear whether other ecosystem functional changes are associated with shifts between picoeukaryotes and prokaryotes. Our analysis is necessarily limited to the spring and summer season and does not consider any form of top-down control (e.g., predation, parasitism) on ecosystem structure. These baseline shifts between picoeukaryotes and prokaryotes provide novel insights into mechanisms supporting the resilience of fjord food webs[49].

Our analysis across picoplanktonic domains revealed that fjord ecosystem structure is dominated by picoeukaryotes in Arctic fjords and prokaryotes in subarctic fjords. Prokaryotic (alpha and beta) diversity was more strongly shaped by temperature gradients, suggesting more substantial changes and increased prokaryotic dispersal into Arctic fjords as temperature will increase in the future. Specifically, oceanographic dispersal of picoplankton communities, a major structuring variable in our analysis, along with a decrease of temperature barriers through increased Atlantification of the Arctic Ocean, can contribute to accelerated change, with additional amplification effects through the expansion of cross-Arctic ship routes and subsequent introduction of alien species[50,51] and/or stimulation of currently dormant cells in Arctic regions. Such taxonomic changes within the base of the food web can imply changes in carbon-cycling efficiency and resilience of existing ecosystem structures. Therefore, microbial diversity and associated functional traits should be monitored across temporal and spatial scales for prokaryotes and eukaryotes simultaneously to identify the risk of biome collapse and establish sustainable solutions for these highly vulnerable ecosystems that are undergoing significant environmental changes.

## Methods
### Sample data and biogeographic regionalization
Data analyzed in this study include datasets from five expeditions in Arctic and subarctic coastal and fjord waters in the Atlantic and Greenlandic sector during spring and summer between 2014 and 2019 (Fig. 1). For broad bioclimatic classifications of sites, we adapted the classifications by the Circumpolar Arctic Vegetation Mapping Project (https://www.arcticcentre.org/EN/arcticregion/Maps/definitions) and grouped sites into the following bioclimatic subzones: high Arctic, low Arctic, subarctic and temperate regions (Supplementary Data 1). The expeditions took place on the research vessels *RV* Maria S. Merian (MSM) and *RV* Heincke (HE). This compiled dataset consists of environmental (temperature, salinity, fluorescence, $PO_4$, $NO_3$, and Si) and picoplankton DNA data from the following cruises: HE431 in northern Norway, southern Norway, and Sweden in 2014; HE492 in Svalbard in 2017, HE533 in northern Norway in 2019; MSM21-3 in

Iceland and west Greenland in 2012 and MSM56 in Svalbard and east Greenland in 2016. Samples were taken along each fjord length, comprising of three to six samples per fjord (Supplementary Data 1). Sampling and processing of all environmental and sequence data were previously published elsewhere[21,24,39,52] or are described in the Supplementary References (HE533). All sample data is publicly available, and cross-links are summarized in Table S5.

### Microbial DNA processing, amplicon sequencing, amplicon sequence analyses, and taxonomic assignment

**16S and 18S rRNA metabarcoding.** Prokaryotic primers (515F–806R)[53] and eukaryotic primers (TA-Reuk454FWD1–TAReukREV3)[54] for all samples were selected in accordance with the Earth Microbiome Project (http://www.earthmicrobiome.org/protocols-and-standards/). The library preparation preceding the sequencing followed standard protocols (16S Metagenomic Sequencing Library Preparation, Illumina, Part #15044223 Rev.B; Illumina Technology).

**16S/18S rRNA sequence processing and taxonomic assignment.** Amplicon sequence variants (ASVs) were obtained by processing the resulting raw paired-end reads with R (R Core Team, 2013) package DADA2 v1.16.0[55] following a modified version of the DADA2 Pipeline Tutorial (v1.8). Processing of 18S rRNA and 16S rRNA gene reads were performed separately. Reads were pre-filtered by length (minLen = 50) and quality (minQ = 2), followed by removal of the primers. The pre-filtered reads were further filtered by the expected length of the amplicon (240–160 bp for 16S rRNA V4 and 270–220 bp for 18S rRNA V4) and quality, for which the maximum number of expected errors (maxEE) was set to 2.7 for forward reads and 2.2 for reverse reads. De-replication, error learning, and sample inference were then performed on the filtered reads. To obtain the full denoised sequences, the paired-end reads were merged with a minimum overlap (minOverlap) of 20 bp. Finally, chimeras were removed, and the ASV tables were built. Prokaryotic and eukaryotic taxonomy were assigned against the Silva v132 database (Supplementary Data 2) and the pr2 database v4.12 (Supplementary Data 3), respectively. We removed all ASVs annotated as mitochondria or chloroplasts from the 16 S rRNA ASV tables, and ASVs annotated as metazoans from the 18S rRNA ASV table.

**Trophic classifications.** We classified eukaryotic taxa based on literature assignment as either autotroph, mixotroph, heterotroph, or unknown if no information of trophy was available, mainly using previously published trophic annotation tables[56,57]. For prokaryotes, we used a previously published trait database[58] followed by additional literature research, classifying taxa into autotroph (lithoautotroph and photoautotroph), mixotroph, heterotroph, or unknown if no information was available. See Supplementary Data 4 for detailed classifications. Prokaryotic functional diversity is well studied using functional inference of prokaryotic ASVs, including a normalization step based on individual rRNA gene copy numbers to predict the functional abundances, which then balances the biases of microorganisms with greater gene copy numbers[59]. Therefore, we checked how well our classification aligns with curated functional annotation, i.e., predicted KEGG pathways in PICRUST2[26]. We performed an additional functional assignment of prokaryotic ASVs using PICRUST2, followed by extracting all KO numbers involved in autotrophic energy metabolism (Supplementary Data 5) and mapped their gene abundance against taxonomic abundances of ASVs, which were classified as autotroph based on our literature research.

### Oceanographic dispersal
Numerical simulations were performed using the output of a 3D, high-resolution, general ocean circulation model of the Arctic Mediterranean to compute large ensembles of Lagrangian drifter trajectories to assess the dispersal of microorganisms and map oceanographic connectivity between the sampling sites. We focused on connectivity between fjords, removing sites within fjords to reduce noise created by geographically closely connected sites within fjords. Therefore, for each fjord, sites closest to the fjord mouth were chosen as "representative samples" of fjords in the available model realization.

The hydrodynamic model used to represent the ocean currents in the study area was based on metROMS (https://doi.org/10.5281/zenodo.290667), which couples the state-of-the-art Regional Ocean Modeling System (ROMS, http://myroms.org), a free-surface, hydrostatic, primitive equation ocean general circulation model[60], with the comprehensive dynamic-thermodynamic sea ice model CICE (https://zenodo.org/record/1205674). metROMS was run with a horizontal resolution of $4 \times 4$ km in an orthogonal, curvilinear grid covering the entire Arctic Mediterranean over 2005–2017, referred to as the "A4-setup"[61,62]. A4s initial state and boundary conditions were derived from monthly-averaged global reanalyses[63] and additional forcing along its open boundaries using the global TPXO tidal model[64]. Surface atmospheric forcing was provided from 6-h ERA-Interim reanalysis[63], with additional freshwater sources from major rivers along the coastal boundaries, including glacial runoff from Svalbard and Greenland. Output from the A4-setup contained velocity fields in 32 terrains following vertical layers and a temporal resolution of 24 h. The first two model years were discarded as spin-up, and validation against available observations confirmed that the model satisfactorily reproduced the general features of currents, hydrography, and sea ice.

The hydrodynamic simulations satisfactorily reproduce major advection pathways in the Arctic Mediterranean[65] and synthetic floats from 23 stations located at fjord mouths were introduced to A4 and tracked using TRACMASS (www.tracmass.org[66];) while they were freely advected by the model's daily averaged velocity field. Cohorts of approximately 1,000 floats were seeded every 10 days, evenly distributed between 5 m and 20 m depth in an area of $3 \times 3$ model grid cells centered on each sampling site and tracked over a lifetime of 5 years. To assess the statistical dispersal due to time-varying ocean currents for each sampling site, time series of ensemble-mean drifter concentrations were computed on the A4 grid by counting the number of drifters of a given age per grid cell from all cohorts, divided by the total number of drifters of that age and all cohorts within the model domain (Fig. S19). Based on these calculations, the matrices of oceanographic connectivity between release and receiving sampling sites were obtained (Figs. S8–S13) by averaging the drifter concentration of a given release site within a $15 \times 15$ cell area centered on the receiving site. Station HE533.F02.28A (Tanafjord) and HE431.F02.19 (Sognefjord) were excluded from this analysis, as these stations were located close to land and could not be resolved by the A4 coastline geometry. Because drifter counts generally decrease exponentially as drifters disperse away from their origin, a logarithmic scale was used to compare drifter concentrations.

### Statistics and reproducibility
We performed all data analysis using R v4.2.2 (R Core Team, 2021) and RStudio v2022.12.0 + 353 (Rstudio Team, 2022).

We removed all ASVs with ≤1 instances for eukaryotes and prokaryotes, and performed Bayesian-multiplicative treatments of zeros using the cmultRepl() function of the zComposition package (v1.4.0.1) accounting for sample sparsity and undersampling. Due to the compositional nature of sequence-based diversity analyses[67], we performed CLR-transformation of the ASV tables for redundancy analysis (RDA). Further, we calculated the Aitchison distance of the ASV tables for distance-based analysis. The Aitchison distance is the Euclidean distance of the CLR-transformed samples and thus deals well with data subsetting or aggregation and allows exploring the true linear relationship between samples (Supplementary_Code/data_transformations).

Hill numbers for alpha diversity (sample richness and Shannon entropy)[68] were calculated using the iNEXT package (v3.0.0) with 100 iterations. Pielou evenness was calculated by dividing the Shannon index by sample richness. Differences in alpha diversity scores between bioclimatic subzones were tested with ANOVA for sample variances, followed by a two-sample t-test comparing individual bioclimatic subzones using the stats

package (v4.2.2) (Table S3). *P* values were adjusted for multiple testing using the Bonferroni adjustment (Supplementary_Code/alpha_div).

We screened for associations between hydrodynamic connectivity and microbial beta diversity changes with correlation analysis. Specifically, we turned the oceanographic connectivity matrices for each temporal bin into hydrodynamic distance matrices by calculating the inverse of the log10 of the synthetic particles (i.e., the inverse of the oceanographic connectivity) between each site pair. The total number of observed site pairs was 210. To test the effect of oceanographic connection on microbial beta diversity, we removed sites that were not oceanographically connected from the analysis. We then scaled the data to fall within the values of 0 and 1, with values close to "0" indicating small hydrodynamic distance and values close to "1" indicating large hydrodynamic distance. We calculated the Pearson correlations between the normalized hydrodynamic distances and 16S rRNA and 18S rRNA Aitchison distances using the cor.test() function of the stats package (v4.2.2) (Supplementary_Code/analysis_distance). Further, we calculated the linear regressions between 16S rRNA and 18S rRNA Aitchison distances and temperature differences of sites within and between Bioclimatic subzones.

We used a stepwise permutational analysis approach to identify explanatory variables that are statistically important for microbial beta diversity structure (ordiR2step() function with perm.max set to 200), and investigated the relationship between microbial beta diversity and contextual environmental data using RDA (Supplementary_Code/analysis_multivariate). Residuals of RDAs were checked for normal distribution (Fig. S20). We tested differences between the diversity at our sampled sites (represented by ASV tables associated with bioclimatic subzones, geographic regions, and fjords) with a permutational ANOVA (PERMANOVA) with 999 permutations using the adonis2() function in vegan (v2.6.4). Further, we tested the variance captured by all significant explanatory variables individually with PERMANOVA (Table S2). Environmental variables were checked for normal distribution and z-scored for scale-independent intercomparability. Differences between bioclimatic subzones were tested with the non-parametric Kruskal-Wallis test.

We investigated the relative contribution of trophic functional groups in Arctic, subarctic, and temperate regions of eukaryotic and prokaryotic ASV tables that were independently CLR-transformed and concatenated. The concatenated CLR-transformed ASV tables were then scaled to the sample size. We imported ASV tables, contextual metadata, and taxonomy tables into a phyloseq object using the phyloseq package (v1.36.0) (Supplementary_Code/analysis_network). We tested differences of trophic functional groups of prokaryotes and eukaryotes between bioclimatic subzones with ANOVA for sample variances followed by a two sample t-test (var.equal = FALSE) using the stats package (v4.2.2).

### Reporting summary
Further information on research design is available in the Nature Portfolio Reporting Summary linked to this article.

## Data availability
Sequence data for this study have been deposited in the European Nucleotide Archive (ENA) at EMBL-EBI under accession numbers PRJEB50596 (MSM56), PRJEB50593 (MS21-3), PRJEB50592 (HE431), PRJEB49358 (HE492), PRJEB50059 (HE533), using the brokerage service of the German Federation for Biological Data [GFBio[70]] in compliance with the Minimal Information about any (X) Sequence (MIxS) standard[71]. Contextual metadata is available on PANGAEA under the following dois and summarised in Table S5: https://doi.org/10.1594/PANGAEA.903511 (HE533), https://doi.org/10.1594/PANGAEA.928451 (HE533), https://doi.org/10.1594/PANGAEA.863438 (HE431), https://doi.org/10.1594/PANGAEA.928449 (HE431), https://doi.org/10.1594/PANGAEA.881306 (HE492), https://doi.org/10.1594/PANGAEA.928449 (HE492), https://doi.org/10.1594/PANGAEA.871015 (MSM56), https://doi.org/10.3389/fmars.2019.00412 (MSM56), https://doi.org/10.1594/PANGAEA.819731 (MSM21-3) and https://doi.org/10.1594/PANGAEA.897293 (MSM21-3).

## Code availability
All code is archived and publicly available in github AGJohnAWI/Arctic-Picos under the release v2.1 (https://doi.org/10.5281/zenodo.5781578).

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

## Acknowledgements
We acknowledge previous sampling efforts and processing of samples collected during the HE533, HE492, HE431, MSM56, and MSM21-3 sampling campaigns. This study was supported through the POF IV Research Programme subtopic 6.4 of the Alfred Wegener Institute, Helmholtz Centre for Polar and Marine Research, Germany. PLB was supported by the Alfred Wegener Institute's Frontiers in Arctic Marine Monitoring (FRAM) Programme and the Helmholtz Metadata Collaboration. TH was supported by the Research Council of Norway (grant nos. 332635 and 314826). AMW acknowledges the Ocean Frontier Institute's (Dalhousie) funding from the Canada First Research Excellence Fund (CFREF). We thank Nancy Kühne for her technical support in sample processing. We thank Matthias Ullrich, Eric J. Raes and Niels Fuchs for their comments on this study. We acknowledge the work of the Leibniz Institute on Aging (FLI) in Jena, Germany, for eukaryotic amplicon sequencing. A previous version of this manuscript has been published as part of CH's PhD thesis[69].

## Author contributions
Conceptualization: C.H., U.J. Methodology- individual DNA extraction: P.T. Methodology- amplicon sequence processing: I.M., C.H. Methodology- statistical analysis: C.H. Methodology- hydrodynamic dispersal analysis: T.H. Investigation: C.H., U.J., A.M.W. Visualization: C.H., T.H. Supervision: U.J., P.L.B., A.M.W. Writing- original draft: C.H. Writing- review and editing: C.H., T.H., P.T., I.M., P.L.B., A.M.W., U.J.

## Funding

## Competing interests
The authors declare no competing interests.
