## [Peer Review File · Communications Biology]

Reviewers' comments:

Reviewer #1 (Remarks to the Author):

Leveraging forecasted lagrangian particles trajectories in a high-resolution hydrodynamic model, Horstmann show connectivity of coastal sites across the polar North Atlantic. These sites spanned nutrient, salinity, and biogeographic gradients and are connected by the major advective currents in the region. The lagrangian particles represented microbial consortia, communities of picoeukaryotes and prokaryotes determined by genomic sequencing. The authors used community diversity and dispersal distance metrics to show that ocean temperature is a major determinant of the species composition for prokaryotes, but this effect is reduced at higher sea temperature sites.

The paper is well-written and flows well, building a strong case for future predictions of microbial community shifts as ocean fronts shift poleward. While these claims have been described before (cite articles), the combination of in situ community composition and lagrangian particle-tracking in a model is a unique combination and a major strength of the article, especially over such a large region in the North Atlantic. For this reason, it will be of interest to others in the field, for example larval dispersal of higher trophic groups interested in mechanisms which shape temporal and spatial marine organism niches. One criticism I have is that, while there is discussion of bottom-up controls on diversity, the authors do not mention how grazing and top-down controls could also select for the difference found in the prokaryotic communities. While this may be an important process contributing to the variability you observe, it is beyond the scope of the inferences drawn from your data. It would also be good to discuss the importance of seed-populations which remain dormant in the non-productive season but seek refugia and do not require lateral dispersal mechanisms. I am also concerned that the model does not have accurate boundary conditions within fjords without meltwater, which affect stratification and retention (re-circulation) dynamics. The resolution of the model was stated as not being able to resolve finer scale features within fjords.

There is not much more needed to improve the manuscript, however, it would be nice for there to be an explanation as to why they chose certain statistical analyses. For example, why was the Aitchinson test for sample distances used as opposed to other available tests? This should be explicitly stated for all non-statisticians.

Specific comments are made below:

94: this is an incomplete sentence

97: cascading effects “on” the trophic structure; change

97: “whole” change to “entire”

140: capitalize the section heading

162: what about meteoric inputs?

241: rephrase; I think we would expect different water masses to have different nutrient inventories especially depending on ice-ocean interactions.

247: I think this was a helpful paragraph to provide context for the model. Thank you

281: depth [comma] temperature?

283: Geomorphologically speaking, this is an extremely important observation, that marine-terminating glaciers have greater dissolved Si concentrations. I think this result should be emphasized in the context of silicifying organisms.

288: “warmer and nutrient-rich” indicates an Atlantic source? This should be explicitly stated as it helps

with your point later for migration of Atlantic waters to your more polar coastal sites.

Reviewer #2 (Remarks to the Author):

Water temperature largely limits the expansion of prokaryotes into functionally diverse Arctic fjord picoplankton communities

The paper “ Water temperature largely limits the expansion of prokaryotes into functionally diverse Arctic fjord picoplankton communities” looks at the microbial across 21 fjords. The paper makes use of a very valuable dataset of prokaryote community in several fjords. However in general, I feel the paper lacks a clear structure and the main message and title is not strongly corroborated with the data. Before performing statistics on the data, it would in my view be very interesting to describe the data in different fjords first (Figure 3), give a general overview of the patterns and maybe focus on some in-fjord details (fe on seasonity) as well (How does the community looks in different fjords, how is abundance of some groups different in different places?) Important information is also missing concerning the samples, where in the fjords where they collected? In the head, central part or mouth? In what seasons?

This uncertainty is also relevant within the ocean model. Considering fjords are not resolved in large scale model, I assume the tracers are released at the mouth of the fjords? This is however not specified in the text. However, in-fjord circulation is very important, tracer studies have shown it can take a long time before particles exit the fjord, so a prokaryote cell in head of the fjord while likely take a long time reaching the fjord mouth, impacting the conclusion of this study.

Minor comments

L 47: Arctic fjords are among the most productive regions of the ocean’s biosphere.

This statement is not unreferenced. Fjord are quite productive, but I think it is maybe quite bold statement to call them the most productive in ocean and Arctic

L51: Arctic fjords are characterized 51 by steep spatial and temporal environmental gradients, including strong seasonality

L58-59: This sentence is mentioned a bit randomly in the text without much context.

L 454- 455: This is part of the discussion not conclusion

Reviewer #3 (Remarks to the Author):

The manuscript by Hörstmann et al focuses on the influence of oceanographic conditions as driven

factors on marine microorganisms of the different domains of life in a large geographic region of high latitudes fjord as susceptible environments of climate change.

I found the coherence of the manuscript hard to follow. The title doesn't reflect clearly the paper content. The researchers did an excellent job in analyzing oceanographic shifts using particle modeling to identify the connectivity in the study area, its hydrodynamics and beta-diversity. However, for me is not clear the "conectivity" over the influence of more local variables explaining the heterogeneity found such as nutrients variability. Maybe I missed how the conectivity and for example the biogeochemical dynamics relates including temperature trends.

The main weakness of the manuscript is associated with the classification of functional groups that potentially could affect their inference. Since relevant functional groups like chemolithoautotrophic ammonia oxidizers, such as Thaumarchaeota and Nitrosomonas, Nitrite oxidizers (Nitrospinae), besides other C1 Methane and Methyl oxidizers, which are wrongly classified as heterotrophic. This could potentially impact the results of the "trophic" characterization. Line (140-143). Metabolic prediction should be generated with unbiased analyses using for example curated pipelines such as Tax4fun and others (Inferring microbiota functions from taxonomic genes: a review - PMC (nih.gov)). Results 297 – 305 must be reanalyzed considering the potential influence of chemoautotrophic communities. In addition, I found that the conclusions are highly speculative in the implications of their results specially associated with the functional effects.

Reviewers' comments: (Responses marked in red)

Reviewer #1 (Remarks to the Author):

1) Leveraging forecasted lagrangian particles trajectories in a high-resolution hydrodynamic model, Horstmann show connectivity of coastal sites across the polar North Atlantic. These sites spanned nutrient, salinity, and biogeographic gradients and are connected by the major advective currents in the region. The lagrangian particles represented microbial consortia, communities of picoeukaryotes and prokaryotes determined by genomic sequencing. The authors used community diversity and dispersal distance metrics to show that ocean temperature is a major determinant of the species composition for prokaryotes, but this effect is reduced at higher sea temperature sites.

The paper is well-written and flows well, building a strong case for future predictions of microbial community shifts as ocean fronts shift poleward. While these claims have been described before (cite articles), the combination of in situ community composition and lagrangian particle-tracking in a model is a unique combination and a major strength of the article, especially over such a large region in the North Atlantic. For this reason, it will be of interest to others in the field, for example larval dispersal of higher trophic groups interested in mechanisms which shape temporal and spatial marine organism niches. One criticism I have is that, while there is discussion of bottom-up controls on diversity, the authors do not mention how grazing and top-down controls could also select for the difference found in the prokaryotic communities. While this may be an important process contributing to the variability you observe, it is beyond the scope of the inferences drawn from your data. It would also be good to discuss the importance of seed-populations which remain dormant in the non-productive season but seek refugia and do not require lateral dispersal mechanisms.

Response: Thank you. We agree that these mechanisms may be important factors that should be further investigated, however, this information cannot be inferred from our data. Therefore, we've added the following statement to our discussion:

“This highlights the limitation of our current knowledge on these rapidly changing ecosystems and the urgency to study them more multidisciplinary including the effect of biological interactions (grazing, symbiosis) that span across size classes, but was beyond the scope of our study.” (L. 447-450) and again in L. 511-514: “It remains unclear whether other ecosystem functional changes are associated with shifts between picoeukaryotes and prokaryotes. Our analysis is necessarily limited to the spring and summer season and does not consider any form of top-down control (e.g., predation, parasitism) on ecosystem structure.”

Further, we believe that it could be helpful to note research on this topic and added two references to our discussion: A review by Lennon and Jones (2011) where they discuss the ecological importance of microbial seed banks, and a recent publication by Don Anderson et al., where they found a large amount of dormant *Alexandrium catenella* cysts in the Arctic that could potentially cause massive HABs in the Arctic. We added the following text to our discussion to point out the relevance of these processes:

“However, prokaryotic dispersal could increase in the future due to increasing Atlantification of the Arctic Ocean ⁴⁴, with even more rapid ecological changes through local temperature-driven stimulation of dormant stages (Lennon & Jones 2011; Anderson et al. 2021).” (L. 431-433)

2) I am also concerned that the model does not have accurate boundary conditions within fjords without meltwater, which affect stratification and retention (re-circulation) dynamics.

Response: The hydrodynamic model does indeed include boundary conditions for meltwater runoff in Svalbard and Greenland fjords, as well as runoff from rivers along the coastal boundaries. We amended the model description, explaining the use of atmospheric forcing in combination with coastal river and glacial runoff:

"Surface atmospheric forcing was provided from 6-h ERA-Interim reanalysis, with additional freshwater sources from major rivers along the coastal boundaries, including glacial runoff from Svalbard and Greenland." (L. 179-181)

3) The resolution of the model was stated as not being able to resolve finer scale features within fjords.

Response: We agree with the reviewer that internal fjord dynamics that are not resolved by the hydrodynamic model may impact the picoplankton distribution within the fjords. Acknowledging that this is not the focus of our analysis, we are still confident that the model provides a good representation of the connectivity between fjords. We added several statements to clarify the focus of the dispersal analysis on connectivity between the sites:

"we related changes in picoplankton beta diversity to modeled oceanographic connectivity at the basin scale." (L. 98/99)

"We focused on connectivity between fjords, removing sites within fjords to reduce noise created by geographically closely connected sites within fjords. Therefore, for each fjord, sites closest to the fjord mouth were chosen as "representative samples" of fjords." (L. 167-169)

4) There is not much more needed to improve the manuscript, however, it would be nice for there to be an explanation as to why they chose certain statistical analyses. For example, why was the Aitchison test for sample distances used as opposed to other available tests? This should be explicitly stated for all non-statisticians.

Response: We added more detailed descriptions and references to the text:

Rationale for CLR transformation: "accounting for sample sparsity and undersampling. Due the compositional nature of sequence-based diversity analyses (Gloor et al. 2017)" (L. 209-211)

Rationale for Aitchison distance: "The Aitchison distance is the Euclidean distance of the CLR-transformed samples and thus deals well with data subsetting or aggregation, and allows to explore the true linear relationship between samples" (L. 212-214)

5) Specific comments are made below:

94: this is an incomplete sentence

Response: Thank you. We corrected this mistake to: "We expect different regionally constrained signals for prokaryotes and picoeukaryotes, following the assumption that prokaryotes and picoeukaryotes being differently constrained in their local selection by, for example, temperature²⁰, light availability²¹, and nutrient input dynamics²²." (L. 99-102)

97: cascading effects "on" the trophic structure; change
Corrected

97: "whole" change to "entire"
Corrected

140: capitalize the section heading
Corrected

162: what about meteoric inputs?

Response: See the earlier reply, we amended the model description, explaining the use of atmospheric forcing in combination with coastal river runoff:

"Surface atmospheric forcing was provided from 6-h ERA-Interim reanalysis, with additional freshwater sources from major rivers along the coastal boundaries, including glacial runoff from Svalbard and Greenland." (L. 179-181)

241: rephrase; I think we would expect different water masses to have different nutrient inventories especially depending on ice-ocean interactions.

Response: Thank you. We agree that "despite" was not the correct word choice. We rephrased this sentence to: "We found significant differences in water mass characteristics between bioclimatic subzones (Kruskal-Wallis test, $p < 0.001$), but also high regional variability in dissolved inorganic nutrient (NO_3 , PO_4 , and Si) concentrations (Fig. 1b), primarily due to relatively higher nutrient concentrations in the fjord heads in comparison to other samples in the fjord central part and mouths (Table S1)." (L. 269-274). We agree that the nutrient inventories strongly depend on ice-ocean interaction but would like to point out that our sampling occurred during the ice-free season and measuring the exact effect of sea ice on nutrient inventories was unfortunately beyond the scope of our study.

247: I think this was a helpful paragraph to provide context for the model. Thank you
Thank you for this feedback, we agree that this is an important note to make.

281: depth [comma] temperature?
Corrected

283: Geomorphologically speaking, this is an extremely important observation, that marine-terminating glaciers have greater dissolved Si concentrations. I think this result should be emphasized in the context of silicifying organisms.

Response: Thank you. We agree that this observation is particularly important for silicifying organisms, and we want to point out that we've noted this in our discussion "Arctic picoeukaryotic beta diversity was positively associated with Si concentration, a key element for diatoms, which concentrations will change with glacial melt and increased acidification in Arctic bioclimatic subzones, resulting in cascading effects on the microbial community (Cantoni et al. 2020)." (L. 484-486)

288: "warmer and nutrient-rich" indicates an Atlantic source? This should be explicitly stated as it helps with your point later for migration of Atlantic waters to your more polar coastal sites.

Response: Thank you, this is a very good point. We added this information to the text as follows: L. 321-324 "Sites belonging to temperate regions, and thus more influenced by Atlantic water which is generally considered warmer and nutrient-rich, were strongly associated with temperature, NO_3 and PO_4 concentrations."

Reviewer #2 (Remarks to the Author):

1) Water temperature largely limits the expansion of prokaryotes into functionally diverse Arctic fjord picoplankton communities

The paper “ Water temperature largely limits the expansion of prokaryotes into functionally diverse Arctic fjord picoplankton communities” looks at the microbial across 21 fjords. The paper makes use of a very valuable dataset of prokaryote community in several fjords. However in general, I feel the paper lacks a clear structure and the main message and title is not strongly corroborated with the data. Before performing statistics on the data, it would in my view be very interesting to describe the data in different fjords first (Figure 3), give a general overview of the patterns and maybe focus on some in-fjord details (fe on seasonality) as well (How does the community looks in different fjords, how is abundance of some groups different in different places?) Important information is also missing concerning the samples, where in the fjords where they collected? In the head, central part or mouth? In what seasons?

Response: We thank the reviewer for their comments. We would like to point out that we intentionally focus on broader geographic scales only and small-scale dynamics have already been discussed in other studies (Elferink et al 2017, 2020; Supraha et al. 2022) (also noted in our Material and Method section L. 122-123). We had all information on individual sampling locations in the previous version of our manuscript, but perhaps this wasn’t stated clear enough in our manuscript. To support a better understanding of sampling and individual fjords we added supplementary figures of sampling sites in each geographic region (Fig. S9- S14). We noted our sampling design in our method section before, but repeated this information in the result section in the revised version of our manuscript:

“Three to six samples were taken along the length of each fjord (Fig. S9 – S14). Between fjords, the samples were clustered together in six geographically distinct regions: northern Norway, Sweden/southern Norway, Svalbard, Iceland, East Greenland, and West Greenland (Fig. 1a, Table S1).” (L. 261 -262). Additionally, we would like to point the reviewer to our Supplementary Table 1, which gives an overview of contextual environmental data of each sample.

After consideration, we changed the title of our study to: “Biogeographic gradients of picoplankton diversity indicate increasing dominance of prokaryotes in warmer Arctic fjords”. We believe this title strongly corroborates with our study as it reflects our method and most important outcome of our study.

2) This uncertainty is also relevant within the ocean model. Considering fjords are not resolved in large scale model, I assume the tracers are released at the mouth of the fjords? This is however not specified in the text. However, in-fjord circulation is very important, tracer studies have shown it can take a long time before particles exit the fjord, so a prokaryote cell in head of the fjord while likely take a long time reaching the fjord mouth, impacting the conclusion of this study.

Response: As also explained in the reply to reviewer one, we fully acknowledge that the hydrodynamic model results are not suitable to assess the dispersal *within* the fjords, but rather provide information about the connectivity *between* fjords. We added several statements to clarify the focus of the dispersal analysis on connectivity between the sites on the basin-scale. In addition, we followed the reviewer’s suggestion, specifying that synthetic drifters were releases at the mouth of the fjords:

“Synthetic floats from 23 stations located at fjord mouths were introduced to A4 and tracked using TRACMASS (www.tracmass.org).” (L. 186-189)

3) Minor comments

L 47: Arctic fjords are among the most productive regions of the ocean's biosphere.

This statement is not unreferenced. Fjord are quite productive, but I think it is maybe quite bold statement to call them the most productive in ocean and Arctic

Response: We changed it to "Arctic fjords are among the most productive high-latitude regions of the ocean's biosphere, sustaining important fisheries." (Meire et al. 2017) (L. 50/51)

L51: Arctic fjords are characterized 51 by steep spatial and temporal environmental gradients, including strong seasonality

Response: Apologies, but unfortunately it is not clear to us what issue should be addressed in the above comment.

L58-59: This sentence is mentioned a bit randomly in the text without much context.

Response: We thank the reviewer for this comment. Re-reading the manuscript we agree and we changed the paragraph as follows: "They form unique, regional communities in sub-Antarctic fjords (Maturana-Martínez et al. 2021), whose occurrence and/or maintenance is of high interest in understanding the coupling between biodiversity and physical dynamics of Arctic ecosystems (Schlegel et al. 2023)." (L62-65)

L 454- 455: This is part of the discussion not conclusion

Response: We have moved this sentence to the discussion (now: L. 504/505)

Reviewer #3 (Remarks to the Author):

The manuscript by Hörstmann et al focuses on the influence of oceanographic conditions as driven factors on marine microorganisms of the different domains of life in a large geographic region of high latitudes fjord as susceptible environments of climate change.

1) I found the coherence of the manuscript hard to follow. The title doesn't reflect clearly the paper content.

Response: We thank the reviewer for this remark, and we reconsidered our title and changed it to "Biogeographic gradients of picoplankton diversity indicate increasing dominance of prokaryotes in warmer Arctic fjords" We believe that this supports our study and our key results well.

2) The researchers did an excellent job in analyzing oceanographic shifts using particle modeling to identify the connectivity in the study area, its hydrodynamics and beta-diversity. However, for me is not clear the "conectivity" over the influence of more local variables explaining the heterogeneity found such as nutrients variability. Maybe I missed how the conectivity and for example the biogeochemical dynamics relates including temperature trends.

Response: We thank the reviewer for making this valuable point. Re-reading our manuscript we understand that we jumped one key analysis step which is the extraction of temperature as the most important variable among environmental variables. This association was stronger than oceanographic connectivity. This shows that the effect of temperature is beyond simply being a water mass indicator. We added this additional analysis to our manuscript:

Methods: "Further, we tested the variance captured by all significant explanatory variables individually with PERMANOVA (Table S6)." (L.244-245)

Results: "Among these variables, temperature stood out as the most important structuring environmental variable (Table S6) that also aligned with the first (and most explanatory) RDA axis in both eukaryotes and prokaryotes." (L.314-316) and "Notably, temperature captured two times more of the variance for prokaryotes than for eukaryotes (PERMANOVA, 38% for prokaryotes vs. 19% for eukaryotes, Table S6). Collapsing beta diversity signals to differences in temperature only, we found that Aitchinson distances correlated with temperature differences (Δ temperature) between sites for eukaryotes and prokaryotes (Fig.4 c,d). Prokaryotic Aitchinson distance was significantly correlated with differences in temperature between all bioclimatic subzones, while eukaryotic Aitchinson distance was significantly correlated but confidence intervals were wide with temperature differences, especially when crossing from temperate into subarctic regions (Table S7)." (324-333). We have also changed the order of the occurring figures and hope this improved the flow of our manuscript.

We have discussed these results, and revised our Abstract accordingly.

Discussion: "These regions were also distinct in their environmental conditions, such as nutrient profiles, with temperature being the most important structuring variable between sites. As differences in temperature will become weaker under continuing Atlantification of the Arctic Ocean, reduced temperature barriers can support more prokaryotic expansion into Arctic regions" (L. 405-408) [...] "Prokaryotes were more influenced by hydrodynamic distance than were eukaryotes, suggesting that prokaryotes were more driven by geographic dispersal¹². However, prokaryotic beta diversity was even stronger associated with temperature differences between sites, that could not only be explained by temperature being a water mass indicator, suggesting ecological temperature barriers for prokaryotes."(L.423-425).

Abstract: "Modeled oceanographic connectivity between fjords suggested that transport alone would create a smooth gradient in beta diversity largely following the North Atlantic Current and East Greenland Current. Deviations from this suggested that 1) picoeukaryotes had some strong regional patterns in beta diversity that reduced the effect of oceanographic connectivity, while 2) prokaryotes

were mainly stopped in their dispersal if strong temperature differences between sites were present. Fjords located in high Arctic regions had also generally very low prokaryotic alpha diversity.” (L.37-40)

3) The main weakness of the manuscript is associated with the classification of functional groups that potentially could affect their inference. Since relevant functional groups like chemolithoautotrophic ammonia oxidizers, such as Thaumarchaeota and Nitrosomonas, Nitrite oxidizers (Nitrospinae), besides other C1 Methane and Methyl oxidizers, which are wrongly classified as heterotrophic. This could potentially impact the results of the “trophic” characterization. Line (140-143). Metabolic prediction should be generated with unbiased analyses using for example curated pipelines such as Tax4fun and others (Inferring microbiota functions from taxonomic genes: a review - PMC (nih.gov)). Results 297 – 305 must be reanalyzed considering the potential influence of chemoautotrophic communities.

Response: We re-classified the prokaryotes into trophic functional groups (autotrophs containing now photoautotrophs and chemolithotrophs; mixotrophs; heterotrophs; unknown). We note that direct functional annotation allows a general functional inference but no direct taxonomy-functional profiling (trait-based). Furthermore, unfortunately, functional inference approaches such as PICRUST2 or Tax4Fun2 are not (yet) available for eukaryotic diversity. Therefore, we used our trait-based approach which allows us to cross-compare between eukaryotic and prokaryotic functional groups. To be able to compare prokaryotic and eukaryotic functional trophic traits we continued using the trait-based approach with the following additions:

- 1) We did a primary annotation with another previously published trait database (BactoTraits; <https://doi.org/10.1016/j.ecolind.2021.108047>)
- 2) We did a functional annotation of prokaryotic sequences using PICRUST2. Our rationale for using Picrust2 was that the algorithm works well in predicting poorly characterized taxa compared to other approaches. Further, the documentation is very transparent and reproducible, while Tax4Fun2 (as suggested by the reviewer) has no active supported github by the author (see e.g., this issue on the web: <https://www.biostars.org/p/9527549/>)

We then compared the functional annotation of KOs derived from PICRUST2 with our trait-based functional association. We used “autotrophy” as an example and derived a very good fit between the two methods. (see Figure 1 below). This result gives us confidence that our trait-based classification can be used as relative abundances of trophic functional groups. We have also added this additional analysis to our manuscript as we believe it can be helpful to the community to easily assess the presented functional annotation.

Figure 1. ASV abundance of prokaryotes annotated as “autotroph” based on literature research mapped against functional inference derived from PICRUST2 including all KOs involved in autotrophy.

Method: “we used a previous published trait database³¹ followed by an additional literature research, classifying taxa into autotroph (lithoautotroph and photoautotroph), mixotroph, heterotroph or unknown if no information was available. See Table S3 for detailed classifications. Prokaryotic functional diversity is well studied using functional inference of prokaryotic ASVs including a normalization step based on individual rRNA gene copy numbers to predict the functional abundances, which then balances the biases of microorganisms with greater gene copy numbers³². We therefore checked how well our classification aligns with curated functional annotation, i.e., predicted KEGG pathways in PICRUST2³³. We performed an additional functional assignment of prokaryotic ASVs using PICRUST2, followed by extracting all KO numbers involved in autotrophic energy metabolism (Table S4) and mapped their gene abundance against taxonomic abundances of ASVs which were classified as autotroph based on our literature research.” (L. 151-161)

Results: L.342-351: “We tested the maintenance of trophic functions across bioclimatic subzones by trophic trait assignment, which infer relative contributions of individual trophic groups based on ASV abundance. Notably, trait-based analysis does not account for variability of absolute gene copy numbers per cell, which is normalized in functional inference algorithms such as PICRUST2³³. However, this relationship is currently incompletely understood for eukaryotes. Therefore, we applied trait-based analysis for intercomparability between prokaryotes and eukaryotes and mapped, as an example for the intercomparability between the two approaches, our trophic annotation against the functional inference of prokaryotes for “autotrophy” including photoautotrophs and chemolithoautotrophs. ASV abundance of autotrophic prokaryotes correlated well with gene copy numbers involved in autotrophy derived from PICRUST2 (Fig. S11).”

Discussion: L. 626- 628 “Our functional trait-based analysis allows a comparative functional analyses of prokaryotes and eukaryotes, extending our functional understanding derived from metagenome and functional inference predictions ⁶¹ across domains.”

Apart from this additional analysis and the re-analysis, we also updated our prokaryotic functional analysis and figure (see Figure 2 below; Figure 4 in the revised manuscript) accordingly but would like to highlight that the main outcomes of our manuscript did not change. We still see a clear dominance of functionally diverse picoeukaryotes in Arctic regions and more prokaryotes in subarctic and temperate regions. Interestingly, the functions per se do not change much along the biogeographic regions, which we also additionally highlighted in our discussion “Notably, the ratios between functional groups (auto-, mixo-, heterotroph) remained constant throughout all samples, but the fraction of prokaryotes and picoeukaryotes within these functional groups changed significantly between regions. This suggests that picoeukaryotes and prokaryotes can serve the same function within an ecosystem, a form of functional redundancy that actually occurs between domains. It remains unclear whether other ecosystem functional changes are associated with shifts between picoeukaryotes and prokaryotes.” (L.507-513)

Figure 2. Relative contribution of trophic functional groups of picoeukaryotes and prokaryotes of Arctic, subarctic and temperate regions. Annotated taxa are summarized in Table S3. CLR-transformed ASV tables of picoeukaryotes and prokaryotes were merged and normalized to 1. **b.** sum of each trophic functional group within bioclimatic subzones showing medians, upper and lower hinges, whiskers and outliers. For each trophic functional groups, the boxplots are ordered from top to bottom: Arctic – subarctic – temperate.

4)In addition, I found that the conclusions are highly speculative in the implications of their results specially associated with the functional effects.

Response: We thank the reviewer. We would like to note that we removed speculative phrases from our discussion and conclusion. Additionally, we highlighted the limitations of our study in our discussion “It remains unclear whether other ecosystem functional changes are associated with shifts between picoeukaryotes and prokaryotes. Our analysis is necessarily limited to the spring and summer season and does not consider any form of top-down control (e.g., predation, parasitism) on ecosystem structure.” (L.511-513). We still think it is important to note that the baseline shifts we observed in our study can have cascading effects on ecosystem functions but our results also indicate mechanisms for resilience and functional redundancy across domains.

Reviewers' comments:

Reviewer #1 (Remarks to the Author):

The authors made an excellent effort to respond to the feedback and criticisms on the manuscript. I remain in support of publication of this work, but I would encourage the authors to think critically about fine-scale dynamics within fjords (as I mentioned in the first review). Fjord exchange with the open ocean waters can be quite limited depending on bathymetry, or confined to episodic pulses with strong seasonality. To suggest that the fjord mouth is representative of the rest of the fjord is an overstatement, and the authors should acknowledge this caveat. While I don't have any more changes to suggest, one could also consider the importance of microbes which are dispersed by megafauna during the summer foraging season. For example, eukaryotic phytoplankton are known to attach themselves to the backs of whales.

I support the article for publication.

Excellent job and I am looking forward to the eventual publication.

Reviewer #2 (Remarks to the Author):

- Editor's note: Comments to the editor that their comments appear addressed.

Reviewer #3 (Remarks to the Author):

The reviewed version of the manuscript by Hörstmann et al were significantly improved in its clarity, coherence and depth considering functional inferences and oceanographic variables.

I still have some concerns and specific comments highlighted below.

In addition, I think that the authors should also include at least a general mention of the specific composition, what are the major taxa found? The biology behind this tremendous work should be highlighted, not only related with the trophic groups inferences and indexes changes, but also the identification of specific microbial taxa found at different taxonomic levels.

I have few specific comments.

Abstract

Line 34-35, only mention functions associated with picoeukaryotes since with the data is not possible to indicate that prokaryotes were favored or that the first group were unfavored in temperate fjords. Same in the discussion section 400 -402. Despite normalization, metabarcoding are not a quantitative approach to compare among both domains, the results associated with both and potential predominance in terms of the ecosystem function should be addressed with care. Compare within each domain and mention a potential shift of both functional groups in the

Line 41 - 43, conclusion should be more carefully stated considering previous comment. Besides fisheries implications are highly speculative considering that both domains are part of picoplankton, could the shifts mentioned influence the way carbon is recycled? Is there enough evidence to sustain this, I am not convinced considering only the use of metabarcoding.

Introduction

Line 80 - 82, add reference to this statement.

Results

317, 313, check correct nutrient nomenclature for nitrate and phosphate

Discussion

406 – 408, prokaryotic expansion is mentioned because of "Atlantification", I think that this should be paraphrased, for example, by mentioning that picoeukaryotes associated with photoautotrophic metabolism were unfavored compared with prokaryotes in the picoplankton fraction. The potential advection or/and higher contribution of heterotrophic prokaryotes should be stated separately. 438-441, I think this section should also be discussed carefully considering "the emerging role of prokaryotes", having a higher detection of prokaryotes does not necessarily indicate a higher role of prokaryotes, few taxa within an ecosystem could play a significant function for example in the case of rare taxa in nitrogen cycle. Moreover, prokaryotes in general play an important role always, therefore the paragraph needs to be clarified. If a specific functional group such as photoautotrophs and the way carbon is recycled, then it could be relevant. I suggest the authors to address more clearly the microbial loop and how the changes observed could have an impact.

Conclusion

519, Co-occurrence analyses could be misinterpreted by network analysis, which was not done here.

Reviewers' comments:

Reviewer #1 (Remarks to the Author):

The authors made an excellent effort to respond to the feedback and criticisms on the manuscript. I remain in support of publication of this work, but I would encourage the authors to think critically about fine-scale dynamics within fjords (as I mentioned in the first review). Fjord exchange with the open ocean waters can be quite limited depending on bathymetry, or confined to episodic pulses with strong seasonality. To suggest that the fjord mouth is representative of the rest of the fjord is an overstatement, and the authors should acknowledge this caveat.

Author response: We thank the reviewer, we do not imply that the fjord mouth locations in the model are fully representative of the properties inside all regions of the fjord, but that those give the best representation of the fjord conditions in the available model realization, which also captures the large-scale connectivity. Hence, we augmented the text below to clarify this aspect in the manuscript.:

"We focused on connectivity between fjords, removing sites within fjords to reduce noise created by geographically closely connected sites within fjords. Therefore, for each fjord, sites closest to the fjord mouth were chosen as "representative samples" of fjords **in the available model realization**." (L.173)

While I don't have any more changes to suggest, one could also consider the importance of microbes which are dispersed by megafauna during the summer foraging season. For example, eukaryotic phytoplankton are known to attach themselves to the backs of whales.

Author response: Thank you, this is a very good point but would require additional (megafauna) samples to compare their microbiome to the adjacent water. While this would be an exciting follow-up study it is beyond our study. Acknowledging this dispersal mechanism, we added the following to the discussion: "to extend future studies by including other ecological variables, such as the effect of biological interactions (grazing, symbiosis, **biological dispersal**) that span across size classes ⁵⁵." (L. 450)

I support the article for publication.

Excellent job and I am looking forward to the eventual publication.

Reviewer #2 (Remarks to the Author):

- Editor's note: Comments to the editor that their comments appear addressed.

Reviewer #3 (Remarks to the Author):

The reviewed version of the manuscript by Hörstmann et al were significantly improved in its clarity, coherence and depth considering functional inferences and oceanographic variables. I still have some concerns and specific comments highlighted below.

In addition, I think that the authors should also include at least a general mention of the specific composition, what are the major taxa found? The biology behind this tremendous work should be highlighted, not only related with the trophic groups inferences and indexes changes, but also the identification of specific microbial taxa found at different taxonomic levels.

Author response: We actively choose not to go into detail of the taxonomic groups at this analysis was done for most cruises individually already. This is mainly because the taxonomic differences mainly arise through variations at lower taxonomic ranks or even within species, in the form of different ecotypes. We added this observation to our result and discussion.

Result: "Despite changes in alpha and beta diversity across regions, relative abundances of major taxonomic groups did not majorly change at the Order level across bioclimatic subzones (Fig. S11)." (L.342-244)

Discussion: "We demonstrated that picoplankton beta diversity separated statistically into high Arctic, low Arctic, subarctic, and temperate fjords, arising from different relative abundances at the ASV level, which were, however, not reflected at higher taxonomic ranks (Order level or higher)." (L.410-413)

And added Figure S11 to the Supplementary.

Further, we think that it may be useful for others to have the taxonomic tables available and have added them to the method section (Table S4, S5)

I have few specific comments.

Abstract

Line 34-35, only mention functions associated with picoeukaryotes since with the data is not possible to indicate that prokaryotes were favored or that the first group were unfavored in temperate fjords.

Author response: We corrected this statement to: “Across 21 fjords, we found that Arctic fjords had proportionally more trophically diverse (autotrophic, mixotrophic, and heterotrophic) picoeukaryotes, while subarctic and temperate fjords had relatively more diverse prokaryotic trophic groups.”

Same in the discussion section 400 -402. Despite normalization, metabarcoding are not a quantitative approach to compare among both domains, the results associated with both and potential predominance in terms of the ecosystem function should be addressed with care. Compare within each domain and mention a potential shift of both functional groups in the

Author response: We agree with the reviewer that metabarcoding is not a quantitative approach and needs to be treated with care. However, the CLR-normalization does allow a comparison of relative abundances across domains, see Tipton et al. 2018 for a nice description (<https://doi.org/10.1186/s40168-017-0393-0>). We clarified that we are only referring to relative abundances by including this fact in the revised version “was reflected by low prokaryotic Richness. We documented significantly, proportionally more relative abundances of eukaryotic auto-, mixo- and heterotrophs in Arctic fjords” (L. 400)

Line 41 – 43, conclusion should be more carefully stated considering previous comment. Besides fisheries implications are highly speculative considering that both domains are part of picoplankton, could the shifts mentioned influence the way carbon is recycled? Is there enough evidence to sustain this, I am not convinced considering only the use of metabarcoding.

Author response: We have changed this sentence to “Ultimately, warming of Arctic fjords could induce a fundamental shift from more trophic diverse eukaryotic- to prokaryotic-dominated communities, with profound implications for Arctic ecosystem dynamics including their productivity patterns.” We agree that the implications for future fisheries are highly speculative and have therefore removed this statement. However, picoplankton forms the base of the marine food web and contributes directly to the primary productivity in marine ecosystems. Our trophic analysis shows changes within the functional group depending on their trophic, which also has direct implications for carbon cycling.

Introduction

Line 80 – 82, add reference to this statement.

Author response: We were referring to results by Sunagawa et al. 2015, who identified temperature as the main structuring factor of prokaryotic beta diversity in the global ocean based on the Tara Oceans dataset. We added this reference to the text.

Results

317, 313, check correct nutrient nomenclature for nitrate and phosphate

Author response: corrected.

Discussion

406 – 408, prokaryotic expansion is mentioned because of “Atlantification”, I think that this should be paraphrased, for example, by mentioning that picoeukaryotes associated with photoautotrophic metabolism were unfavored compared with prokaryotes in the picoplankton fraction. The potential advection or/and higher contribution of heterotrophic prokaryotes should be stated separately.

Author response: We were here also referring to the [currently] reduced alpha diversity of prokaryotes in Arctic fjords and relative abundance of prokaryotes vs. picoeukaryotes (see Fig. S12) and not only to the functional groups. We understand that “Atlantification” may be a too broad term here. Further, we agree that only referring to heterotrophic groups should be replaced with a more general term. We therefore rephrased (and added an according reference describing the process of increasing water temperatures in the Arctic) to:

“As temperature differences may decline in a future Arctic Ocean ⁴⁵, reduced temperature barriers can support more prokaryotic expansion into Arctic regions with fundamental ecosystem baseline shifts among trophic functional groups.” (L.405-407)

438-441, I think this section should also be discussed carefully considering “the emerging role of prokaryotes”, having a higher detection of prokaryotes does not necessarily indicate a higher role of prokaryotes, few taxa within an ecosystem could play a significant function for example in the case of rare taxa in nitrogen cycle. Moreover, prokaryotes in general play an important role always, therefore the paragraph needs to be clarified. If a specific functional group such as photoautotrophs and the way carbon is recycled, then it could be relevant. I suggest the authors to address more clearly the microbial loop and how the changes observed could have an impact.

Author response: This paragraph is solely discussing the diversity patterns of prokaryotes and picoeukaryotes. As prokaryotes are more temperature-dependent they may become more and more diverse in a future Arctic Ocean and should therefore be included in ecosystem analysis studies. We acknowledge that “role” may not be fitting in this context as it refers to ecosystem functions which are not discussed in this paragraph. We therefore rephrased this sentence to:

“Our comparative analysis of picoeukaryotic and prokaryotic dispersal patterns highlights a **potentially** emerging **dominance of more diverse** prokaryotes in previously eukaryotic-dominated Arctic ecosystems within the picoplankton size fraction, adding a holistic perspective of future ecosystem changes.” (L. 453-456)

Conclusion

519, Co-occurrence analyses could be misinterpreted by network analysis, which was not done here.

Author response: Rephrased to “Our analysis across picoplanktonic domains [...]” (L. 548)

REVIEWERS' COMMENTS:

Reviewer #4 (Remarks to the Author):

The manuscript is clearly improved, the authors adequately respond to the comments and doubts arisen during the last review.

I recommend its publication in the journal.

Bremerhaven, 06.feb.2024

Dear Dr. Grinham,

Thank you very much for your time and patience with our manuscript. Please find below the last response to the reviewer.

Reviewer 4 (Remarks to the Author):

The manuscript is clearly improved, and the authors adequately respond to the comments and doubts that arose during the last review.

I recommend its publication in the journal.

Author response: We thank the reviewers for their time and comments that helped us to improve our manuscript.